# Dexmedetomidine: An Alternative to Pain Treatment in Neonatology

**DOI:** 10.3390/children10030454

**Published:** 2023-02-25

**Authors:** Laura Mantecón-Fernández, Sonia Lareu-Vidal, Clara González-López, Gonzalo Solís-Sánchez, Marta Suárez-Rodríguez

**Affiliations:** 1Neonatolog Unit, AGC Pediatrics, Hospital Universitario Central de Asturias, 33011 Oviedo, Spain; 2Instituto Investigación Sanitaria Principado de Asturias, ISPA, 33011 Oviedo, Spain; 3Division of Neonatology, Hospital for Sick Children, Department of Pediatrics, University of Toronto, Toronto, ON M5G 1X8, Canada; 4Medical Department, University of Oviedo, 33003 Oviedo, Spain

**Keywords:** neonatal pain, pain management, dexmedetomidine

## Abstract

Infants might be exposed to pain during their admissions in the neonatal intensive care unit [NICU], both from their underlying conditions and several invasive procedures required during their stay. Considering the particularities of this population, recognition and adequate management of pain continues to be a challenge for neonatologists and investigators. Diverse therapies are available for treatment, including non-pharmacological pain management measures and pharmacological agents (sucrose, opioids, midazolam, acetaminophen, topical agents…) and research continues. In recent years one of the most promising drugs for analgesia has been dexmedetomidine, an alpha-2 adrenergic receptor agonist. It has shown a promising efficacy and safety profile as it produces anxiolysis, sedation and analgesia without respiratory depression. Moreover, studies have shown a neuroprotective role in animal models which could be beneficial to neonatal population, especially in preterm newborns. Side effects of this therapy are mainly cardiovascular, but in most studies published, those were not severe and did not require specific therapeutic measures for their resolution. The main objective of this article is to summarize the existing literature on neonatal pain management strategies available and review the efficacy of dexmedetomidine as a new therapy with increasing use in the NICU.

## 1. Introduction

Management in neonatal pain continues to be a challenge for neonatologists and investigators. During the 80′s and 90′s, it was believed that neonates did not feel pain because of the immaturity of their nervous system. These wrong beliefs have influenced the neonatal pain assessment during decades [1]. Only 30 years ago, newborns underwent major surgical procedures without perioperative or postoperative analgesia [2]. Fortunately, further investigations documented the great vulnerability of preterm newborns to the metabolic and clinical effects of pain, hence pain management began to change in this population. As multiple studies have shown, neonates not only suffer pain, but is has short and long-term consequences such as metabolic and hormonal stress, even neurodevelopmental sequellae. There are many concerns about the side effects of pain treatment and there is still so much to study on how to assess neonatal pain and what treatment would be the best in a rapidly developing target organ [1]. 


**KEY POINTS**


-Dexmedetomidine is a selective alpha-dose agonist with sedative and analgesic action.-In clinical studies, it has been shown to be useful in preterm and term neonates under mechanical ventilation, during short invasive procedures and for postoperative pain control.-The main side effects are cardiovascular, mostly mild and self-limited.-Its potential neuroprotective role would make it the treatment of choice in patients undergoing active hypothermia.-The current scientific evidence about its use is weak, so new studies are needed to determine its optimal dose and to prove its efficacy and safety.

## 2. Effects of Repetitive Pain in the Neonate

As mentioned before, scientific evidence demonstrates that neonatal pain has short and long-term adverse effects [1,2,3]. By 23–25 weeks, free nerve endings and spinal cord projections are fully mature, so pain sensation is theoretically possible at 24 weeks of gestation [4]. Neonatal pain, especially in preterms and in those who are exposed to multiple painful procedures, influences brain maturation, reduce white matter integrity and it is related to neuronal loss [2,4]. On the other hand, neonatal pain secondary to surgical procedures has been associated with neurodevelopmental disorders and with lower scores on congnitive tests, results that are maintained during adolescence [5]. Related to this, two recent studies showed a reduced thalamus and amygdala volumes in children under 32 weeks of age that underwent painful invasive procedures [6,7]. Giordano V et al. [8] demonstrated how pain affects neurodevelopment in infants under 32 weeks of age by using Bayley Scales of Infant Development at 12 months of corrected age. They observed lower scores in the Bayley Scales in the group of children who were exposed to more pain procedures.

In addition to the neurodevelopment effects that pain produces in this population, significant pain has also been shown to affect neonatal growth, leading to lower percentiles for body weight and head circumference at 32 weeks of gestational age, regardless of other medical confounding factors [4]. 

Even though these adverse effects and sequellae are already known, it has been described that a patient admitted to a Neonatal Intensive Care Unit (NICU) receives between 7.5 and 17 painful procedures per day [1,4]. Recent changes in NICU, as Newborn Individualized Developmental Care and Assesment program (NIDCAP) or the use of pain assessment scales make it possible to reduce painful procedures. Roofthooft et al. [9] observed in a level 3 NICU that the mean number of painful procedures per patient per day had significantly declined from 14.3 in 2001 to 12.2 in 2009. Recently, studies are being carried out to reduce skin breaks and blood tests in NICU and secondarily, reduce neonatal pain procedures [10,11]. Therefore, it is essential for the well-being and development of these patients to find the appropriate pain scale for its better evaluation and find the best treatment (pharmacological and non-pharmacological) that relive not only pain but also has minimal or no adverse effects at any time. 

## 3. How to Assess Neonatal Pain: Pain Assessment Scales

During hospitalization, the newborn is exposed to many unpleasant painful procedures, including venipuncture, the insertion of peripheral venous catheters, endotracheal intubation, invasive mechanical ventilation, suctioning, therapeutic hypothermia and neonatal surgery among others. Accurate assessment of pain is vital to ensure the optimal effectiveness of pain management therapy in these neonates who experienced pain during their NICU stay. One of the main barriers that hinder optimal pain management in NICU is how health professionals perceive it and, therefore, how they treat it [12]. For these reason, nurses and neonatologists must be well trained in neonatal pain assessment to ensure adequate pain management [13], being essential to recognize signs and symptoms of pain and adequately assess it by using appropriate pain scales [14,15]. There are up to 65 scales to evaluate pain or sedation in children in a preverbal stage of development [15,16]. The American Academy of Pediatrics (AAP) recommends using the following 5 scales to assess neonatal pain (Table 1): Neonatal Facial Coding System (NFCS), Premature Infant Pain Profile-Revised (PIPP-R), Neonatal Pain, Agitation and Sedation Scale (N-PASS), Behavioral Indicators of Infant Pain (BIIP) and Acute Pain in Newborns/Douleur Aiguë du Nouveau-né (APN/DAN) [15,17]. 

Patients admitted to the NICU are diverse and pain scales are validated only after certain gestational age or type of pain, so it is very difficult to implement just one [13]. Olsson et al. [14] observed that 15.6% of the studies in neonatal pain used a pain scale that was either not validated for the type of pain or for the type of patient [13]. 

There is no consensus on which scales should be used for pain assessment and moreover, the perception of pain with these scales is completely subjective. For these reason, new lines of research have emerged looking for a more objective measurement of pain: near-infrared spectroscopy (NIRS), electroencephalogram (EEG), skin conductance and cortisol levels. Regarding the use of NIRS, Slater et al. [18] described how painful episodes produced certain hemodynamic changes in the absence of behavioral ones. Because of this, NIRS could provide a better and an objective measurement of pain, but its implementation is problematic due to certain interferences that can occur owing to movement or other hemodynamic or respiratory changes that may interfere with the displayed value, making its clinical utility questionable [13]. 

EEG measures the activity of cortical neurons, but it is difficult to specify whether this activity is only due to pain or there is interference with movement, especially in preterm neonates. Skin conductance reflects increment in sympathetic nervous system activity but was found to have low sensitivity and specificity for assessing neonatal pain. Blood cortisol levels are a biomarker of pain and stress. These levels cannot be used for pain management in clinical practice because we cannot immediately obtain its value. Finally, it has been suggested that heart rate variation could be an objective indicator of neonatal pain intensity. Nevertheless, some investigators such as Oberlander y Saul found that there are other medical conditions that may interfere with the usefulness of heart rate for the assessment of neonatal pain [19]. 

Recently, in order to have an objective measurement of neonatal pain, it has been introduced into the clinical practice of NICUs the use of Newborn and Infant Parasympathetic Evaluation (NIPE) monitor. This monitor analyzes the heart rate variability and allows to evaluate autonomic nervous system activity [20,21] by giving a value from 0 to 100. It is an objective measurement, easy to evaluate, not observer dependent and no invasive [21,22]. Ivanic et al. [20] observed that NIPE monitor is useful for postoperative pain management. In the same way, Uberos et al. [23] describe that NIPE index is a good method to measure nociceptive stimuli in very low birth weight newborns. 

## 4. Management of Pain

### 4.1. Non-Pharmacological Pain Management

Non-pharmacological pain management are recommended as first line treatment of pain in this population. As some authors have published before, regarding ethical considerations, they are preferred because their minimal risk and great benefit [24,25]. 

Non pharmacological interventions to relief the pain in neonates could be clustered into three categories based on their hypothesized mechanism of action [26]. 

-Environmental strategies: context-focused interventions that modify the environment in which the neonate suffers pain has been showed to reduced pain reactivity. Some examples of these strategies are the following: clustering procedures, minimal touching when appropriate, lowering noise, reducing light intensity, closing gently incubator doors, covering incubator surfaces to decrease the intensity of lights, or just adjusting alarms to a non-stressful level are some strategies that have been shown to be effective.-Cognitive strategies: different strategies that modify the perception of pain in the neonate such as distraction techniques by using toys or videos [26]. It is to note that distraction is not recommended for preterm neonates because of its immature motor and cognitive systems.-Behavioral strategies: these strategies consist of directly or indirectly manipulate the neonate to block the transmission of nociception or directly activate inhibitory pathways [24]. More than fifteen strategies have demonstrated their benefits to relief the pain [27]: skin to skin contact (Kangaroo care), facilitated tucking (hand-hugging), non-nutritive sucking and other sucking strategies (finger or pacifier), swaddle methods, rocking, breastfeeding, co-bedding (placing twins together in the same incubator) [25] sweet solutions, touch-massage therapies, simulated mother’s voice, parents presence, familiar odor, nonfamiliar odor (such as vanilla, lavender or other pleasant smells) [28], warming the neonate’s heel before a heel lance procedure, music therapy or even touching crochet toys (such as octopus) during painful procedures among others [29].

The combination of various of these techniques have been shown to be beneficial and preferable to the use of a single one [27] and should be used not only during but also before the painful procedure. 

### 4.2. Pharmacological Pain Management

-Sucrose/Glucose

Both glucose and sucrose, can be considered pharmacological or non-pharmacological pain therapy. Classically, the most studied molecule and the one that has shown the most effectiveness has been sucrose. The concentration, timing and route of administration differs from the different published articles [24]. 

A minimum effective concentration of 24% sucrose, two minutes before the painful procedure, has been shown to be effective in relieving pain in this population [30,31]. 

-Opioids

The most studied treatments in this group of patients are morphine and fentanyl. Others such as remifentanil do not have long-term effects studies. Both, morphine and fentanyl take a short time to take effect (5 min versus 1–2 min respectively) and are preferrable to use in mechanical ventilated neonates and during therapeutic hypothermia. Morphine is preferable in postoperative pain and there are some concerns about its use in extremely preterm neonates. Some studies have recently demonstrated adverse long-tern neurodevelopmental effects [32] whereas others demonstrated that low dose infusion did not affect [18,33]. These long-term effect do not occur with fentanyl [34]. Regarding mechanical ventilation, it is preferable not to do it routinely, but only in selected patients [18].

-Non-opiods

Midazolam is the most widely used benzodiazepine in the neonatal period. It is used as sedative, as it has little analgesic effect [18]. It has published that midazolam produces short-term adverse effects and, as with some opioids, there is also concerns about long-term neurodevelopmental effects [35,36]. Besides, midazolam is highly associated with delirium, specially in preterm patients and in those undergoing cardiac palliation. 

Acetaminophen [paracetamol] is another alternative for postoperative mild-to moderate pain management [37]. Compared to morphine, no differences were found when using Neonatal Infant Acute Pain Assessment Scale scores. In addition, it should be noted its absence of adverse effects such as liver toxicity and it does not produce withdrawal symptoms as opioids do [38]. 

There are some pharmacokinetics data of acetaminophen in preterm neonates even from 24 weeks [39], but it has been published long-term adverse effects such as neurocognitive impairment like autism spectrum disorders [40,41].

Other non-steroidal anti-inflammatory drugs (NSAIDs) such as indomethacin or ibuprofen are only used to close the patent ductus arteriosus (PDA) in this population. 

Despite ketamine, methadone or propofol are commonly used outside the NICU, further studies are needed in neonates [18,42,43]. 

-Loco-regional techniques

The most studied topical agent is eutectic mixture of local anesthetics (EMLA^®^, 2.5% lidocaine, 2.5% pilocarpine). It has shown safety and effectiveness [18,42,43] in certain minor procedures as in circumcision, where dorsal penile nerve block (DPNB) and EMLA cream are comparably safe and effective. 

Another example of regional anesthesia is the one used during laser treatment of retinopathy prematurity (ROP). Classically, general anesthesia or sedation has been used for its treatment, but topical anesthesia has recently begun to be used (for example with propacaine). However, a recent review has shown its relationship with life-threatening events so more studies are required for its safe use [44]. 

## 5. New pharmacological Treatments: Dexmedetomidine

As previously mentioned, is essential to find new molecules with a good safety profile that have minimal or no side effects [45]. One of the most promising treatments is dexmedetomidine due to both, its efficacy and its safety profile as well as its neuroprotective potential.

### 5.1. Working Mechanism

Dexmedetomidine is a selective alpha-2 receptor agonist with a broad spectrum of pharmacological properties. Its chemical structure is similar to clonidine, but more specific for alpha-2 than alpha-1 receptors (1620:1 versus 220:1 respectively), which makes its use safer in the critically ill patients. Cardiovascular effects are dose-dependent: at lower perfusion rates, central effects dominate by producing a decrease in heart rate and blood pressure and at higher doses, peripheral vasoconstrictor effects prevail, leading to an increase in systemic vascular resistance and blood pressure, while the bradycardic effect is enhanced. 

It produces anxiolysis, sedation and analgesia without causing respiratory depression. The sympatholytic effect is produced by decreased noradrenaline release from sympathetic nerve endings and its sedative effects are mediated by the inhibition of the locus coeruleus, which is the predominant noradrenergic nucleus located in the brainstem. Its analgesic effect is moderate and potentiates the effect of opioids when are used in association [46]. 

### 5.2. Pharmacokinetics and Pharmacodynamics 

A good understanding of pharmacokinetics (PK) and pharmacodynamics is crucial to optimize the use of any drug and limit the occurrence of adverse effects. Moreover, the neonate is assumed to have an immature metabolism, so clearance of many drugs is expected to be lower. Therefore, they will be exposed to greater toxicity. 

Dexmedetomidine is metabolized in the liver by glucuronization and through cytochrome P450 (CYP 2 A6). Most of the metabolites are eliminated by the kidneys and a small part through the feces. 

Specific studies in the neonatal population are scarce [39,47,48]. We can find an interesting phase II/III, open-label multicenter trial study about its safety, efficacy, and PK in preterm and term neonates published by Constantinos et al. in 2014. They enrolled critically ill and mechanically ventilated neonates divided into two age groups (from 28 to 44 weeks of gestational age). They conclude that PK profile of dexmedetomidine appears to be different in neonates compared with older children, with a longer half-live of the drug [49]. 

Therefore, it is to be expected, as it this has been demonstrated in several publications, that its clearance is even lower at a lower gestational age [47,50,51]. In these cases, the mean doses required to achieve the same level of sedation and to avoid adverse effects would be lower than in older patients.

Many of the publications in newborns focus on the patient undergoing active hypothermia or patients undergoing cardiac surgery [52,53]. Drug clearance would also be lower in those cases and the need to administer loading boluses to achieve adequate concentrations in a reasonable time is postulated by Mc Adams et al. [50]. 

The use of intravenous doses between 0.1–2 mcg/kg/hour has been described to be effective, with mean effective dose around 0.3–0.5 mcg/kg/hour (highest in surgical patients). Initial intravenous loading doses range from 0.5 to 1 mcg/kg/hour. When the selected route is intranasal, in patients undergoing short procedures, doses between 2.5 and 4 mcg/kg/hour are described [54,55]. 

### 5.3. Clinical uses of Dexmedetomidine: Present and Future

In 1999 the United States Food and Drug Administration (FDA) approved its use for intravenous route for up to 24 h in adults admitted to intensive care unit and in 2003 it was additionally approved for procedural sedation [56]. However, many other routes of administration are being explored in clinical practice and it seems to be a molecule with a great potential in many clinical situations. Data obtained from experimental animals and adult population have encouraged its off-label use [57] also in pediatric and neonatal patients. Consequently, numerous studies have been published about this subject in recent years, but the degree of evidence is moderate, being mostly retrospective observational studies. 

### 5.4. Anesthesia and Postoperative Pain

Dexmedetomidine was used for the first time in cardiac surgery, both during anesthesia and postoperative care period. Zimmerman et al. [51] conducted a cohort study in children under 36 weeks. They found that maintenance doses between 0.2 and 2 mcg/kg/h were effective and safe. They also studied pharmacokinetics and demonstrated significantly decreasing drug clearance during extracorporeal circulation. On the one hand, it seems to reduce the use of other anesthetics with deleterious long-term neurological effects [58] as it has been proved by Zhou X et al. [59] in animal experimentation studies. On the other hand, dexmedetomidine seems to attenuate neuroendocrine damage and the hemodynamic response to trauma from surgery and extracorporeal circulation [60]. Subsequently, its use was extended to other types of surgery, with similar advantages and a good safety profile in the pediatric patient [61,62]. 

### 5.5. Specific Uses in Neonatology 

More specifically in the field of neonatology, its use has recently been described by Ojha S et al. and Tauzin et al. in two patient-focused reviews on invasive and non invasive mechanical ventilation [63,64]. It has been shown to be safe and as effective as fentanyl in premature infants when administered as continuous intravenous infusion. According to some studies, among its advantages, we could find a shorter duration of additional sedation-analgesia as well as a decrease in the total dose of opioids and benzodiazepines administered [65,66,67]. In both cases, it would mean a reduction in the side effects of classic drugs, a shorter duration of mechanical ventilation and a faster total enteral intake, which would be equivalent to more rapid removal of central catheters and the consequent lower risk of sepsis.

Sperotto et al. focused on evaluating the prevalence of withdrawal syndrome and delirium in patients undergoing prolonged sedoanalgesia (more than 24 h) in a multicenter observational study [68]. The association of dexmedetomidine with classic treatments seems to significantly decrease the score on the commonly used pain withdrawal scales (Cornell, Withdrawal Assessment Tool-1) in addition to providing adequate sedation and reducing the doses of other drugs [benzodiazepines, opioids, propofol or ketamine]. Another secondary benefit in these patients would be a more rapid withdrawal of respiratory support. 

In summary, its use in term and preterm neonate appears to be useful, not only because of the intrinsic properties of dexmedetomidine as a sedative and analgesic, but also by reducing the use of benzodiazepines and opioids. This allows the side effects of these drugs to be reduced in both, short and long term.

Occasionally, its use as specific treatment for withdrawal syndrome once it has already been established has also been described [69].

Another potential benefit that has been widely studied is neuroprotection, which makes this treatment an ideal option for the neonatal patient, especially preterm neonates, who are the most vulnerable to the neurodevelopmental effects of the molecules that are traditionally used in sedoanalgesia [70,71,72]. This neuroprotective function has been proven in several preclinical studies of injury models including ischemia-reperfusion, inflammation, and traumatic brain injury as well as adult clinical trials of brain trauma [73,74].

Experimental studies in animals have demonstrated the absence of histopathological changes in their brain, kidneys and liver when dexmedetomidine is used in anesthesia in combination with other treatments, making it a safe and an effective option. Furthermore, it does not seem to cause cognitive dysfunction in treated animals. Among other mechanisms of action, it seems to increase the expression of protein kinases 1 and 2 in the hippocampus, which are mediators of neuronal survival and synaptic plasticity [52]. In preclinical hypoxia-ischemia studies, it suppresses cytokine-mediated brain damage, reducing tissue loss and improving neurological function. 

In clinical studies in patients undergoing active hypothermia, its efficacy has been demonstrated when comparing to morphine, as well as the reduced need for other treatments to control pain and/or agitation [75]. Besides, it does not seem to have an impact on respiratory drive, cardiovascular complications are rare and transitory, and it does not affect the tolerance of enteral feeding. On the other hand, unlike other molecules, dexmedetomidine does not alter the electroencephalogram pattern, whose interpretation is crucial in the first hours of life of these patients. 

### 5.6. Safety Profile 

Regarding its safety profile, its side effects are mainly cardiovascular. Sperotto et al. [68] found 37% of cardiovascular side effects among treated patients and O’Mara [76] found no significant alterations in blood pressure in a sample of non-surgical patients. Dersch et al. [77] found a higher number of side effects in a retrospective review in a cohort of neonates undergoing surgical procedures admitted to their unit (41.7% of hypotension and 69.4% of bradycardia). It is important to note that in this study, the patients received prolonged infusions of dexmedetomidine [with a mean duration of 11 days] and most of them were concomitantly receiving opioids. Furthermore, the post-surgical state may also be a factor in the development of hypotension. 

In most of the studies, these events are not serious and although they lead to a reduction or treatment withdrawal, they do not require specific therapeutic measures for their resolution [68,77,78]. It is also worth mentioning that the vast majority of the patients were concomitantly receiving benzodiazepines and/or opioids. This is not a consistent finding, but in many studies, it seems to be associated with the administration of a loading dose of dexmedetomidine at the beginning of the treatment and the use of higher doses during the treatment, above 1 mcg/kg/hour.

Another issue to highlight is the occurrence of withdrawal symptoms after prolonged exposure. Dersch et al. [77] described the occurrence of at least one withdrawal-related symptom in 70% of their cohort. In contrast, O Mara et al. [76] in a case-control study in neonates connected to mechanical ventilation did not diagnose any withdrawal syndrome in the group of patients treated with dexmedetomidine. These discrepancies may be due to the absence of specific withdrawal scales, the unspecificity of withdrawal symptoms [mainly tachycardia and hypertension] and, once again, the concomitant use of opioids. This is another important point to clarify in order to establish recommendations for weaning. 

## 6. Conclusions

In conclusion, when administered intravenously, treatment with dexmedetomidine seems to be an effective and a safe alternative for pain treatment and sedation in the neonatal patient. The intranasal route could be an alternative for short procedures. There are several indications in which its use in neonatology is becoming commonplace: patients connected to mechanical ventilation, short painful invasive procedures and during postoperative period. On the other side, its neuroprotective potential makes it a promising molecule in the treatment of patients with neurological damage, especially during active hypothermia. Nevertheless, for the moment there are no studies that analyze neurological long-term development of children exposed to dexmedetomidine in the neonatal period.

It would be desirable to expand pharmacokinetic studies to optimize the recommended doses and established randomized clinical trials to verify its efficacy and safety for each of the above-mentioned indications. 

## Figures and Tables

**Table 1 children-10-00454-t001:** Neonatal Pain Scales for different target populations [13].

Scale	Construct	Age	Type of Scale
BIIP	Acute pain	24–32 GW	Behavioral/multivariable
PIPP-R	Acute pain	26–40 GW	Multidimensional/multivariable
NFCS	Ventilated child, prolonged pain, postoperative pain	29 GW to 18 months	Behavioral/multivariable
N-PASS	Acute pain, prolonged pain, sedation	23–30 GW	Multidimensional/multivariable
APN/DAN	Acute pain	25–41 GW	Behavioral/multivariable

Abreviations: BIIP: Behavioral Indicators of Infant Pain, PIPP-R: premature Infant Pain Profile-Revised, NFCS: Neonatal Facial Coding System, N-PASS: Neonatal Pain, Agitation and Sedation Scale, APN/DAN: Acute Pain in Newborns/Douleur Aiguë du Nouveau-né, GW: gestational weeks.

## Data Availability

No data were created.

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
