# Peer review of "Dexmedetomidine: An Alternative to Pain Treatment in Neonatology"

_children, 2023, doi:10.3390/children10030454_

Round 1

Reviewer 1 Report

Line 41: Would cite the actual source referred to in Reference 2 not an article that referred to the work by Dr. Anand

Line 92: Need to characterize what these unpleasant procedures are

Line 188: Need to add that midazolam is highly associated with delirium especially in preterm patients and those undergoing cardiac palliation

Disagree that there are only case reports or observational series.  Please use the following as a review of literature

Mahmoud M, Barbi E, Mason KP. Dexmedetomidine: What's New for Pediatrics? A Narrative Review. J Clin Med. 2020 Aug 24;9(9):2724. doi: 10.3390/jcm9092724. PMID: 32846947; PMCID: PMC7565844.

The authors have overlooked to of the main advantages of dexmedetomidine - avoiding the use of benzodiazepines and reduction in opiates which allow earlier extubation in infants undergoing cardiac palliation along with greatly reduced delirium.

Author Response

Dear Mr/Mrs,

First of all, thank you for your review and your comments about the manuscript. We have modified it according to your suggestions. We answer your questions in the following lines:

  • Line 41: Would cite the actual source referred to in Reference 2 not an article that referred to the work by Dr. Anand: we have already made the change, thank you.

  • Line 92: Need to characterize what these unpleasant procedures are: during hospitalization, the newborn is exposed to many unpleasant painful procedures such as venipuncture, the insertion of peripheral venous catheters, endotracheal intubation, invasive mechanical ventilation, suctioning, therapeutic hypothermia, and neonatal surgery among others. We have already added it to the manuscript, thank you four appreciation.

  • Line 188: Need to add that midazolam is highly associated with delirium especially in preterm patients and those undergoing cardiac palliation: thank you, we will add it to the text.

  • Disagree that there are only case reports or observational series.  Please use the following as a review of literature

Mahmoud M, Barbi E, Mason KP. Dexmedetomidine: What's New for Pediatrics? A Narrative Review. J Clin Med. 2020 Aug 24;9(9):2724. doi:10.3390/jcm9092724. PMID: 32846947; PMCID: PMC7565844.

Indeed, larger studies have already been published, although most of them are retrospective and observational.  We take it into account to add it to our review, thank you so much.

  • The authors have overlooked to of the main advantages of dexmedetomidine - avoiding the use of benzodiazepines and reduction in opiates which allow earlier extubation in infants undergoing cardiac palliation along with greatly reduced delirium: effectively, that’s right. We will include it in the text.

Thanks again , your contributions have allowed us to greatly  improve the manuscript. 

Reviewer 2 Report

I have considered this topic relevant and with significant contribution to the field. But a minor revision is necessary to clarify the citations and upgrade the abstract. The suggestions were given in the attached file.

Author Response

Dear Mr/Mrs,

Thank you for your review and your comments about the manuscript. We have modified it in the text according to your suggestions.

We have done an exhaustive review of the text and citations and we have improved the introduction as proposed.

Thanks again , your contributions have allowed us to greatly  improve the manuscript. 

Reviewer 3 Report

Review of the paper: “Dexmedetomidine: an alternative to pain treatment in neonatology.”

Overall this is a very interesting review on a very relevant topic. However, I have several remarks regarding the presentation of the paper as well as on the use of references in this manuscript.

When the authors describe the effects of repetitive pain in the neonate( in paragraph 2), they reference to studies that involve foetuses, preterms and older neonates. It may be an option to move ‘chronologically’ during fetal and postnatal development when they discuss this phenomenon. In their references, they often use other reviews. For instance in the statement about how many painful procedures that a neonate -who has been admitted to NICU- receives, they refer to 2 reviews (ref 1  and 5). Yet the very first study on this topic was published in 2003 (https://pubmed.ncbi.nlm.nih.gov/14609893/ ) and -as a practising physician- I know that recent changes in neonatology have ensured that fewer painful procedures are currently carried out on patients admitted to these units (especially in the developed world). So if they make a statement on this topic, it would be prudent to include early studies and later studies that likely will show a relevant reduction.

‘Reduction of pain’ is a topic that is often used in this manuscript, yet nowhere could I find that the concept of ‘reduction of procedures’. It is logical that a major reduction of pain in neonates would be ensured by a reduction in the number of painful procedures (such as phlebotomy and suctioning of the endotracheal tube or any other painful procedure) to what is clinically acceptable.

They adequately discuss the abundance of neonatal pain scales and the fact that it seems to be impossible to get a ‘common denominator’ or generally accepted scale. Yet in the same paragraph it would also be interesting to find a table with painful procedures and their different levels. E.g. : is a venous puncture more painful than a suctioning of the endotracheal tube? It is clear that procedures that are much more painful should be treated with more potent drugs than procedures that are much less painful.

In paragraph 5 the authors discuss the introduction of dexmedetomidine as a new pharmacological treatment for pain. I will confess that I did not check every single reference in this manuscript, but in this paragraph the authors describe that this drug can even have a neuroprotective effect and reference to (44) and (45). However, reference 45 is -again- a review that states that neuroprotection of this drug has only been studied in rats and not in humans (https://pubmed.ncbi.nlm.nih.gov/28158247/ ). In this instance I think that the reader would be helped by clear communication (stating that this has only been tested in rats and reference to the original study and not a ‘later review’).  Moreover, in reference 45 one can read that ‘no long-term studies in humans for development outcomes have been performed’ with dexmedetomidine.

The entire part about dexmedetomidine (part 5 of this text) which is -as the title suggests- the main topic of this review, is very poorly structured. At present many studies are included but some have been done in adults, some in children and some in neonates. In a review about the use of dexmedetomidine in neonatology this should be stated very clearly. I think that this paragraph may benefit from a balanced ‘substructure’. For instance; the authors could start with the working mechanism and effects of the drug, followed by the current understanding about its pharmacokinetics and pharmacodynamics, this then could lead towards an overview of the preclinical and clinical studies and at the end specific reference to all studies in neonatology.

I have some minor remarks regarding language. There are a number of grammatical and typographical errors in this manuscript. On line 44, the statement ‘It is worldwide known’ is likely written by somebody who is not a native speaker. On line 46 the term ‘sequelaes’ should be ‘sequellae’. And there are numerous other examples.

Author Response

Dear Mr/Mrs,

First of all, thank you for your review and your comments about the manuscript. We have modified it according to your suggestions. We answer your questions in the following lines:

  • When the authors describe the effects of repetitive pain in the neonate (in paragraph 2), they reference to studies that involve foetuses, preterms and older neonates. It may be an option to move ‘chronologically’ during fetal and postnatal development when they discuss this phenomenon: thank you for your suggestion:

To be honest, we just wanted to highlight when pain perception seems to be present, what happens around week 24 of pregnancy. We did not want to make the introduction very extensive, as is not the main topic of the review.

  • In their references, they often use other reviews. For instance in the statement about how many painful procedures that a neonate -who has been admitted to NICU- receives, they refer to 2 reviews (ref 1  and 5). Yet the very first study on this topic was published in 2003 (https://pubmed.ncbi.nlm.nih.gov/14609893/ ) and -as a practising physician- I know that recent changes in neonatology have ensured that fewer painful procedures are currently carried out on patients admitted to these units (especially in the developed world). So if they make a statement on this topic, it would be prudent to include early studies and later studies that likely will show a relevant reduction:

Thank you for your suggestion. The truth is that, even though adverse effects and sequellae of repeated pain are already known, it has been described that a patient admitted to a NICU receives between 7.5 and 17 painful procedures per day [1,5]. Recent changes in NICU, as NIDCAP program or the use of pain assessment scales make it possible to reduce painful procedures. Roofthooft et al observed in a level 3 NICU that the mean number of painful procedures per patient per day had statistically significantly declined from 14.3 in 2001 to 12.2 in 2009. Recently, studies are being carried out to reduce skin breaks and blood tests in NICU and secondarily, reduce neonatal pain procedures (Zhou L et al). 

  1. Roofthooft D.W.E., Simons S.H.P. Anand K.J.S. Tibboel D., van Dijk M. Eight years later, are we still hurting newborn infants? Neonatology. 2014;105:218-226
  2. Zhou L, Taylor J, Kdman A, Stewart A, Bhatia R. Staff awareness and bunding reduce skin breaks and blood tests in neonatal intensive care. J Paediatr Child Health. 2021;57:1485-1489.
  3. Klunk CJ, Barrett RE, Peterec SM, Blythe E, Brockett R, Kenney M, Natusch A, Thursland C, Gallagher PG, Pando R, Bizzarro MJ. An initiative to decrease laboratory testing in a NICU. Pediatrics. 2021;148:e2020000570. Doi: 10.1542.

 We take your comments  into account to improve the quality of the review.

  • ‘Reduction of pain’ is a topic that is often used in this manuscript, yet nowhere could I find that the concept of ‘reduction of procedures’. It is logical that a major reduction of pain in neonates would be ensured by a reduction in the number of painful procedures (such as phlebotomy and suctioning of the endotracheal tube or any other painful procedure) to what is clinically acceptable.

They adequately discuss the abundance of neonatal pain scales and the fact that it seems to be impossible to get a ‘common denominator’ or generally accepted scale. Yet in the same paragraph it would also be interesting to find a table with painful procedures and their different levels. E.g. : is a venous puncture more painful than a suctioning of the endotracheal tube? It is clear that procedures that are much more painful should be treated with more potent drugs than procedures that are much less painful:

Thank you for your comment. We have already included the table that you propose in the new version of the review.

- In paragraph 5 the authors discuss the introduction of dexmedetomidine as a new pharmacological treatment for pain. I will confess that I did not check every single reference in this manuscript, but in this paragraph the authors describe that this drug can even have a neuroprotective effect and reference to (44) and (45). However, reference 45 is -again- a review that states that neuroprotection of this drug has only been studied in rats and not in humans (https://pubmed.ncbi.nlm.nih.gov/28158247/ ). In this instance I think that the reader would be helped by clear communication (stating that this has only been tested in rats and reference to the original study and not a ‘later review’).  Moreover, in reference 45 one can read that ‘no long-term studies in humans for development outcomes have been performed’ with dexmedetomidine: 

Regarding paragraph 5, you are right to point out that there are only preclinical studies about the role of dexmedetomidine on neuroprotection We have therefore modified both, the text and the references.

  • The entire part about dexmedetomidine (part 5 of this text) which is -as the title suggests- the main topic of this review, is very poorly structured. At present many studies are included but some have been done in adults, some in children and some in neonates. In a review about the use of dexmedetomidine in neonatology this should be stated very clearly. I think that this paragraph may benefit from a balanced ‘substructure’. For instance, the authors could start with the working mechanism and effects of the drug, followed by the current understanding about its pharmacokinetics and pharmacodynamics, this then could lead towards an overview of the preclinical and clinical studies and at the end specific reference to all studies in neonatology:

It seemed to us a very good idea to restructure the main part of the review to make it clearly described. We have done it in a slightly different way than you have proposed.

  • I have some minor remarks regarding language. There are a number of grammatical and typographical errors in this manuscript. On line 44, the statement ‘It is worldwide known’ is likely written by somebody who is not a native speaker. On line 46 the term ‘sequelaes’ should be ‘sequellae’. And there are numerous other examples.

As you have suggested, we have made an entire revision of the language. We are not English native speakers, so we hope that the modifications that you propose  have improved language quality.

Thanks again , your contributions have allowed us to greatly  improve the manuscript. 

Round 2

Reviewer 3 Report

The authors have adequately addressed all reviewer comments and suggestions.